# Exploring acculturative stress and coping mechanisms among pregnant South Asian immigrants in Ontario, Canada: A qualitative study protocol

Anam Shahil-Feroz [1]*, Mehtab Jaffer[1], Kateryna Metersky [2], Saleema Allana[1], Salima Meherali [3], Shokoufeh Modanloo[1], Usha George[4], Samya Hasan[5], Zohra Lassi [6,7]

1 Faculty of Health Sciences, Arthur Labatt Family School of Nursing, The University of Western Ontario, London, Ontario, Canada, 2 Daphne Cockwell School of Nursing, Toronto Metropolitan University, Toronto, Ontario, Canada, 3 Faculty of Nursing, University of Alberta, Edmonton, Alberta, Canada, 4 School of Social Work, Toronto Metropolitan University, Toronto, Ontario, Canada, 5 Council of Agencies Serving South Asians, Toronto, Ontario, Canada, 6 Faculty of Health and Medical Sciences, School of Public Health, University of Adelaide, Adelaide, Australia, 7 Robinson Research Institute, University of Adelaide, Adelaide, Australia

* ashahilf@uwo.ca

## Abstract

### Background

Canada's South Asian immigrant population has grown by 154% over the past two decades, making it the largest visible minority group in the country. Many Pregnant South Asian Immigrants (PSAIs) have unique reproductive well-being needs and face challenges adapting to new cultural and social contexts. Despite support from agencies like the Council of Agencies Serving South Asians (CASSA), PSAIs encounter cultural differences, language barriers, limited social support, financial uncertainties, and restricted access to culturally responsive resources. These factors increase acculturative stress, yet there is limited evidence on how PSAIs experience and cope with these stressors.

### Goal and objectives

This one-year project aims to deepen understanding of acculturation challenges faced by PSAIs in Ontario through two objectives: (1) explore PSAIs' experiences of acculturative stress and coping mechanisms, focusing on cultural, social, and structural factors that increase stress during pregnancy, and (2) develop actionable recommendations to inform strategic adaptations of existing programs and services to better meet PSAIs' needs.

**Data availability statement:** No datasets were generated or analysed during the current study. All relevant data from this study will be made available upon study completion.

**Funding:** This research is funded by the Social Sciences and Humanities Research Council Partnership Engage Grant (Application No. 892-2024-3059). The funders had no role in study design, data collection and analysis, decision to publish, or preparation of the manuscript.

**Competing interests:** The authors have declared that no competing interests exist.

## Methods and analysis

A constructivist qualitative design will be used to conduct 25-30 semi-structured interviews with PSAIs in Ontario. Interviews will explore perceptions related to cultural adaptation, language barriers, social support, sources of stress, and coping mechanisms. Data will be analyzed using Braun and Clarke's reflexive thematic analysis and guided by Berry's acculturation framework. The analysis will identify key themes on acculturation, stress, and coping. Data will inform evidence-based recommendations to improve support programs and services.

## Ethics

Ethical approval has been obtained from Western University Health Sciences Research Ethics Board (HSREB) in June 2025 (Protocol # 127113).

## Discussion

This project will enhance understanding of PSAIs' acculturation experiences in Ontario, identify gaps in current supports, explore coping mechanisms, and assess needs. Findings will inform culturally safe interventions and resources and lay the foundation for future interdisciplinary studies focused on co-designing, implementing, and evaluating culturally relevant supports for PSAIs.

## Introduction

Canada is one of the world's most diverse and multicultural nations being home to over 8.3 million immigrants, who make up 23% of its population as of 2021 [1]. Among these, South Asians (SA) —individuals from India, Pakistan, Bangladesh, and Sri Lanka – represent 7.1% of Canada's overall population [2,3]. Over the past two decades (2001–2021), the SA population has grown significantly (154%), outpacing other demographic groups [4]. Predominantly, SA immigrants settle in Ontario, British Columbia, and Alberta [2], making these provinces primary hubs for cultural diversity and immigrant settlement. Many immigrants travel across international borders during their reproductive years in search of better life opportunities [5], with some experiencing their first pregnancy in Canada. While there is a lack of systematic sociodemographic data collection on this group, anecdotal evidence from settlement agencies indicates that Pregnant South Asian Immigrants (PSAI) make up a significant proportion of recent newcomers to Canada.

Pregnancy typically brings joy to pregnant individuals and their families; however, for PSAIs, the journey to motherhood is particularly stressful due to the added complexities of acculturation—a process of cultural and psychological change that results following meeting between cultures [6]. Several factors uniquely shape acculturation experiences for SA immigrants and are documented in the literature as contributors to acculturative stress. For instance, rigid gender roles in many SA communities often place the primary responsibility for childcare and household tasks on mothers,

while fathers serve as primary earners [7–10]. This dynamic can intensify acculturative stress during pregnancy, limiting their time, autonomy, and opportunities for cultural integration. Limited extended family support, a key source of practical and emotional assistance in SA households, further increases women's caregiving burden after migration, as they must balance household responsibilities, employment, and the broader demands of adapting to a new environment [7,11,12]. Language barriers also restrict women's ability to communicate with healthcare providers, navigate services, and build new social connections, all of which are crucial for successful adaptation [7,8]. In addition, loss of familiar social support networks [7,11], economic uncertainties [11,13], experiences of social oppression [7,12] can lead to feelings of isolation, stress, and decreased well-being [14] and erode women's sense of stability and belonging [12]. Finally, limited awareness of culturally safe resources impede access to supportive services that could otherwise facilitate the acculturation process [7,11,14]. Collectively, these factors contribute to acculturative stress [15], which refers to the psychological strain arising from the pressures of adjusting to new cultural and social environments [16]. These stressors often hinder PSAIs' access to essential health and social services, and compromise their ability to assert their well-being needs and preferences during pregnancy [17]. As a result, PSAIs frequently experience feelings of fear, isolation, anxiety, and depression, which can have long-term consequences on both maternal and fetal health and well-being [18]. These challenges can also exacerbate social disparities, restricting access to support and services, and further entrenching their vulnerability within the community [7,11]. Given the limited support available, PSAIs often turn to social media outlets such as Facebook (e.g., Soul Sisters Canada, SA Mommies of Milton, Brown Mommies Canada) [19] and other informal sources for immediate, accessible emotional and psychological support to cope with isolation and acculturative stress.

Existing research on the acculturation process of SA immigrants has predominantly explored areas such as acculturation-related daily hassles and psychological adjustment [20], life satisfaction [11,21], mental health [22], discrimination [12], identity crisis [23,24], overall well-being, and the associations between acculturation and dietary practices [25]. Although current evidence offers valuable insights into acculturation among SA immigrants, there is a significant gap in understanding how PSAIs experience acculturation in Canada. To date, only one study has examined the negative impact of acculturation on PSAIs' health behaviours, social support, and stress in Montreal, Canada [26]. Specifically, higher levels of acculturation were associated with dieting during pregnancy, inadequate social support, and heightened stress levels [26]. This is particularly important because the acculturation challenges faced by PSAIs are significantly intensified by the additional complexities and vulnerabilities associated with pregnancy. For example, PSAIs experience reduced access to familial and community support systems due to language barriers, migration-related separation, and unfamiliarity with local healthcare and social services. Positive acculturation experiences for PSAIs typically involve the availability of culturally safe and personalized social supports [27,28]. In contrast, PSAIs without such supports are at an increased risk of experiencing poor health and well-being outcomes for both themselves and their fetuses [26,29]. Therefore, it is both timely and necessary to understand how PSAIs experience acculturative stress and their coping mechanisms. This knowledge can inform strategic adaptations to existing programs and services, improve access to reliable support networks, and enhance the overall well-being and integration of PSAIs in Canada.

## Aim

The overall aim of this 1-year project is to gain a deeper understanding of the unique acculturation challenges faced by PSAIs in Ontario, Canada, and to refine current programs and services to better serve PSAIs. The overarching research question is: How do PSAIs in Ontario experience acculturative stress, and what coping mechanisms do they employ during their transition to a new cultural environment? This project has two key objectives:

1. To explore PSAIs experiences of acculturative stress and coping mechanisms in Ontario, Canada with a focus on cultural, social, and structural factors that intensify stress during pregnancy.

2. To develop actionable recommendations for adapting existing programs and services to better support PSAIs' reproductive well-being and integration.

## Theoretical underpinning

This project is rooted in an equity-oriented approach, which focuses on identifying and addressing systemic and structural inequities that affect health and social outcomes. Within this equity lens, we will examine the cultural, social, and structural factors that influence PSAIs' access to health and social services for addressing their reproductive well-being needs. This includes uncovering the structural and systemic marginalization that PSAIs face as they navigate the challenges of immigrating to a new country. Our overarching aim is to advocate for the development of equitable, culturally safe support services that enhance the well-being of PSAIs in Ontario, empowering them to cope and thrive in their new cultural environment.

The theoretical perspective guiding this work is Berry's Acculturation Model (1980) [30] (Fig 1), which proposes a two-dimensional framework for understanding acculturation. The first dimension concerns the retention or rejection of an individual's native culture, while the second addresses the adoption or rejection of the host culture. From these dimensions, four acculturation strategies emerge: 1) **Assimilation**, occurs when individuals adopt the cultural norms of a dominant or host culture, over their original culture; 2) **Separation**, occurs when individuals reject the dominant or host culture in favor of preserving their culture of origin; 3) **Integration**, occurs when individuals are able to adopt the cultural norms of the dominant or host culture while maintaining their culture of origin.; and 4) **Marginalization**, occurs when individuals reject both their culture of origin and the dominant host culture. This model also emphasizes the concept of **acculturative stress** [30], which arises from navigating the challenges posed by these cultural intersections. This model will serve as an analytical framework for exploring the experiences of acculturative stress and coping mechanisms among PSAIs in Ontario, Canada. Through this framework, the study aims to uncover how cultural, social, and structural factors, such as cultural differences, language barriers, loss of social support networks, and ethnic discrimination intensify acculturative stress and influence the coping mechanisms employed by this population [21,31].

In the context of SA immigrant communities in Canada, Berry's four acculturation strategies provide a useful conceptual lens for anticipating how acculturation may shape stress experiences. Assimilation pressures, for example, may arise as PSAIs navigate expectations to adopt Western norms surrounding autonomy, gender roles, or parenting practices, which can contrast with collectivist values and extended family-based support systems common in SA households. Conversely, separation may occur when individuals strongly maintain heritage cultural practices, potentially leading to tension when these practices are misunderstood or not accommodated within Canadian social and health systems. Integration, often considered an adaptive strategy, may require structural and social supports that enable PSAIs to balance cultural continuity with participation in the host society, particularly during pregnancy and early postpartum when formal and informal

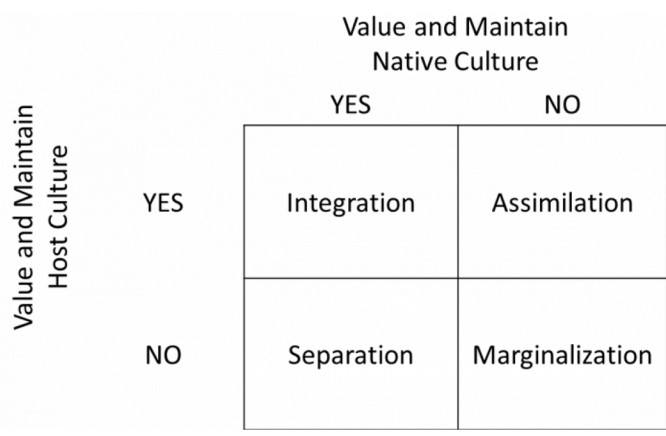

**Fig 1. Berry's Model of Acculturation.**

support networks are critical. Marginalization may emerge in contexts where individuals experience both weakened ties to their heritage culture and barriers to engaging with the host culture, such as discrimination, language challenges, or unfamiliarity with health and social services. These anticipated dynamics illustrate how Berry's model will guide our analytic approach and help identify cultural, social, and structural factors that may influence acculturative stress and coping among PSAIs in Ontario.

## The community partner organization

This project was developed in response to a request from a community organization, Council of Agencies Serving South Asians (CASSA), driven by their recognition of the unique acculturation challenges faced by PSAIs and their families as they adapt to a new cultural environment in Ontario. CASSA identified this critical need through its extensive work with PSAIs and their families, particularly in relation to the immigration and settlement process. CASSA serves as a social justice umbrella organization that supports Ontario's SA communities in addressing a wide range of social, political, economic, and cultural barriers. CASSA operates across Ontario, Alberta, and British Columbia, leading projects and advocating for policies focused on racialized health and well-being, systemic racism, racial equity in education, gender-based violence, and hate eradication. Its mission is to advance social justice and equity for SA communities across Canada. Through its various programs, CASSA engages a diverse array of communities, policymakers, experts, advocates, and academics via advisory tables, steering committees, public forums, consultations, galas, fundraisers, awareness campaigns, and social media. CASSA has played a central role in shaping the project and conceptualizing it alongside the research team, ensuring that it aligns with their strategic priorities, particularly around immigration and settlement support. CASSA will play a key role in the project's activities, including facilitating the recruitment of PSAIs through its extensive network of SA agencies, contributing to the development of interview guides, participating in interviews, and attending analysis meetings to ensure that the evolving analysis remains relevant and useful. They will also help in co-creating knowledge products and disseminating knowledge mobilization (KMob) products across their network of 120 agencies and 12 individual leaders. Throughout the project, CASSA will continue to provide valuable intellectual leadership and engage in shared decision-making.

## Materials and methods

### Study design

This project will employ a constructivist qualitative design, using Braun & Clarke's reflexive thematic analysis [32]. This approach is appropriate because it emphasizes that themes are actively constructed through researcher engagement with the data, rather than passively "discovered," allowing for a nuanced understanding of participants' lived experiences. Reflexivity will be central, as researchers will reflect on their perspectives, assumptions, and experiences throughout the study. Previous research on the phenomenon, acculturative stress among PSAIs [11,20–22], provides a valuable basis for shaping the project's design and setting clear boundaries for investigation. The findings emerging from this approach will accomplish the project's first objective, which is to collect and analyze the experiences of acculturative stress and coping mechanisms among PSAIs in Ontario.

### Study participants and setting

We plan to purposively recruit first-generation PSAIs who self-identify as SA, currently reside in Ontario, and immigrated to Canada within the past five years, as this period represents a critical window in the acculturation process when individuals may experience heightened stress and adaptation challenges [31]. Eligible participants must have experienced at least one pregnancy, whether first or subsequent, after migrating to Canada within the past five years. Participants do not need to be currently pregnant. To explore diverse experiences, we will use a maximum variation sampling strategy. This

will include participants across different stages of pregnancy post-immigration (immediately upon arrival, one to two years, and three to five years after migration), various immigration statuses (Canadian citizens, permanent residents, temporary residents, asylum seekers, and individuals with precarious or undocumented status), employment statuses, countries of origin (Pakistan, India, Bangladesh, and Sri Lanka), and languages spoken (English, Urdu, Punjabi, Hindi, and Bangla). Study participants recruitment will be facilitated by CASSA. If needed, PSAI will also be recruited through other sources such as healthcare providers and social media platforms that cater to SA populations.

## Data collection

We will conduct semi-structured in-depth individual interviews with PSAI to explore their perceptions and experiences related to cultural adaptation, language barriers, cultural differences, social support, sources of stress, and coping mechanisms. Participants will be invited to take part in virtual interviews via a separate project email. Each interview will follow a semi-structured guide [33] (See Appendix A in S1 File) that includes open-ended questions. The guide has been developed iteratively based on a review of the literature on acculturative stress and perinatal experiences among SA immigrant populations, and informed by Berry's acculturation framework (1980). The guide has been reviewed by the study team and will be pilot tested with 2–3 individuals to assess clarity, cultural relevance, and interview flow. Revisions from the pilot testing will be incorporated, and the final interview guide will be included in the subsequent manuscript reporting the study findings. The interviews will be conducted by the Principal Investigator (PI) and Graduate Research Assistant (GRA) via Zoom (Zoom Video Communications, Inc.). The decision to conduct interviews via Zoom was guided by considerations of accessibility, flexibility, and participant comfort. Given that participants may have caregiving responsibilities, transportation barriers, or geographic dispersion, virtual interviews reduce logistical burden and may facilitate participation, particularly among immigrant populations. Interviews will be conducted in the participants' preferred language—Hindi, Urdu, English, Punjabi, or Bangla. The PI is proficient in English, Hindi, and Urdu, and we will hire a GRA with proficiency in at least one SA language, preferably Bangla or Punjabi. If necessary, we will engage a professional interpreter for Punjabi and Bangla interviews. We plan to conduct approximately 25–30 semi-structured interviews, with each lasting between 40 and 60 minutes. The final number of interviews will be guided by the concept of information power [34] that is, the more relevant and rich the information provided by participants in relation to the study aim, the fewer participants are required. We will also prioritize depth and richness to capture diverse experiences based on immigration status, pregnancy timeframe post-arrival, language spoken, and country of origin. Interviews will be simultaneously transcribed and then translated into English for analysis. Participants will receive an honorarium (20 CAD per hour) recognizing the value of their time.

## Data analysis

Data will be analyzed using Braun and Clarke's reflexive thematic analysis [32], which emphasizes the active role of the researcher in constructing themes. Anonymized interview transcripts will be imported into NVivo (version 12 Plus; QSR International) [35]. The PI and GRA will engage in an iterative, inductive process, beginning with data familiarization, generating initial codes, identifying themes, and refining these themes through ongoing analytic reflection. Coding will be reflexive and interpretive. The PI and GRA will actively reflect on their perspectives, assumptions, and experiences, acknowledging how these shape the construction of themes. Reflexive journaling and regular analytic discussions will support transparency, consider alternative interpretations, and enhance analytic rigor. Themes will be constructed to capture both explicit and latent meanings in participants' accounts, with particular attention to experiences of acculturative stress, integration, and assimilation, interpreted through Berry's (1980) framework [30]. The analysis will identify patterns, similarities, and differences in reproductive well-being needs, barriers to adapting to a new cultural environment during pregnancy, navigation of social services, and coping strategies.

## Study timeline

Participant recruitment is currently ongoing and commenced in November 2025. Recruitment is expected to continue until June 2026. Data collection will occur concurrently with recruitment and is anticipated to be completed by July–August 2026. Data analysis will begin in August–September 2026, with completion of analysis and reporting of results expected by December 2026.

## Ethics

Ethical approval for this project has been obtained from the Western University Health Sciences Research Ethics Board (HSREB) in June 2025 (Protocol # 127113). All eligible participants will receive detailed information about the study, including its objectives, potential risks and benefits, procedures, the nature of their involvement, and their rights such as the ability to withdraw at any time without consequence. Only authorized members of the research team will have access to study data. All study data will be securely stored for up to seven years following the completion of the study, in accordance with institutional policies.

## Discussion

Canada relies heavily on immigration to fuel population and economic growth, with SAs constituting the largest visible minority group in the country. Challenges faced by PSAIs in integrating into their communities can adversely affect Canada's statutory commitment to multiculturalism and influence PSAIs' well-being. This project will significantly enhance the understanding of the acculturation experiences of PSAIs in Ontario. It will identify gaps in the current support available through settlement agencies, explore coping mechanisms, and assess the needs of PSAIs. These findings will inform the development of practical recommendations for community organizations and settlement agencies in Ontario to facilitate successful integration for PSAIs and their families, ensuring a more inclusive society and promoting better opportunities and outcomes for PSAIs and their offspring. It will also inform the development of evidence-based, culturally safe interventions and resources to better serve this group. In the long run, the project will also strengthen a research partnership with CASSA and PSAIs, laying the foundation for future interdisciplinary and collaborative studies focused on co-designing, implementing, and evaluating culturally relevant resources for PSAIs.

As part of the dissemination of knowledge, we will publish a peer-reviewed article in an open-access journal, such as the Journal of International Migration and Integration. Additionally, the findings will be presented at key conferences, including Pathways to Prosperity, with a focus on equity, social justice, and immigrant well-being. To disseminate the knowledge outside academic audiences, we will develop and share a plain language report with CASSA and other community organizations with the intention of providing information around the specific challenges PSAIs face during the acculturation process and emphasize the urgent need for improved support systems through culturally safe resources. To ensure frontline service providers are well-equipped to meet the unique needs of PSAIs, we will conduct targeted training sessions for service providers. These events will not only educate staff about the unique needs of PSAIs but will also provide practical recommendations for enhancing their support through available services. To further empower PSAIs and their families, the findings will be communicated through accessible formats, including voice messages and animated videos in Urdu, Hindi, Punjabi, and Bangla. These resources aim to equip PSAIs and their families with the knowledge and tools necessary to advocate for changes in current systems and contribute to the co-design of culturally safe resources.

## Limitations

As this manuscript describes a study protocol, findings are not yet available. Potential limitations include the possibility of recruitment challenges due to language diversity and varying immigration statuses, which may affect the representativeness of the sample. The study is also limited to PSAIs residing in Ontario, which may reduce generalizability to other

provinces or South Asian immigrant populations in Canada. Additionally, as interviews will be conducted virtually, we acknowledge potential limitations, including challenges in establishing rapport, reduced capacity to observe non-verbal communication, technological barriers (e.g., limited internet access or digital literacy), and privacy concerns within shared living environments. To address these limitations, we will provide technical assistance as needed and facilitate referrals to organizations that offer access to digital devices and internet connectivity, including libraries and community partners (CASSA). Despite these limitations, the study's purposive and maximum variation sampling approach is designed to capture diverse perspectives to ensure rich and meaningful insights.

## Conclusion

This study protocol outlines a qualitative investigation into acculturative stress and coping mechanisms among PSAIs in Ontario, Canada. The project aims to enhance understanding of cultural, social, and structural factors that influence acculturation experiences during pregnancy, and to inform the development of culturally safe, evidence-based interventions. Findings will provide actionable recommendations for community organizations, support agencies, and policymakers, while establishing a foundation for future collaborative research focused on improving reproductive well-being and integration for PSAIs.

## Supporting information

**S1 File. Interview Guide.**
(DOCX)

## Author contributions

**Conceptualization:** Anam Shahil-Feroz, Salima Meherali, Shokoufeh Modanloo, Usha George, Zohra Lassi.

**Funding acquisition:** Anam Shahil-Feroz, Kateryna Metersky, Saleema Allana, Salima Meherali, Usha George, Samya Hasan, Zohra Lassi.

**Investigation:** Anam Shahil-Feroz, Saleema Allana, Salima Meherali, Shokoufeh Modanloo.

**Methodology:** Anam Shahil-Feroz, Kateryna Metersky, Salima Meherali, Shokoufeh Modanloo, Usha George, Zohra Lassi.

**Project administration:** Anam Shahil-Feroz, Mehtab Jaffer, Saleema Allana, Samya Hasan.

**Resources:** Anam Shahil-Feroz, Samya Hasan.

**Supervision:** Salima Meherali, Usha George, Zohra Lassi.

**Writing – original draft:** Anam Shahil-Feroz.

**Writing – review & editing:** Anam Shahil-Feroz, Kateryna Metersky, Saleema Allana, Salima Meherali, Shokoufeh Modanloo, Usha George, Zohra Lassi.

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
