## [Editor Report · Decision Letter 0]

10 Feb 2026

Dear Dr. Shahil-Feroz,

Thank you for submitting your manuscript to PLOS ONE. After careful consideration, we feel that it has merit but does not fully meet PLOS ONE’s publication criteria as it currently stands. Therefore, we invite you to submit a revised version of the manuscript that addresses the points raised during the review process.

The protocol contains all relevant information, the study is important and the methodology sound. There are only a few details that the authors should address before acceptance:

In the Financial Disclosure, please specify if the funders played any role in the study design, data collection, analysis, decision to publish or preparation of the manuscript.According to the title, the aim is to explore acculturative stress and “coping mechanisms”, then in the aims “coping strategies” are mentioned. Are mechanisms and strategies the same?In the Introduction, acculturation is described as “a process where individuals strive to preserve their cultural identity while adapting to Canadian values and norms”. But acculturation doesn’t refer exclusively to Canadian values. Please provide a more general description of this construct.The interviews will be conducted over zoom. What was the rationale for choosing this over in-person interviewing? Could you discuss the limitations that this form of interviewing implies?Please specify the currency of the honorarium to be given to participants.In the Data Analysis section, the reference for thematic analysis is Braun & Clark. But those authors’ reflexive approach is different from the interpretive description proposed by Thorne. Please specify which “flavor” of thematic analysis will be used for this project.

We look forward to receiving your revised manuscript.

Kind regards,

Ietza Bojorquez, Ph.D.

Academic Editor

PLOS One

Journal Requirements:

“This research is funded by the Social Sciences and Humanities Research Council Partnership Engage Grant (Application No. 892-2024-3059). “

---

## [Author Response · Author response to Decision Letter 1]

24 Feb 2026

COVER LETTER

Dear Ietza Bojorquez,

PLOS One

We are very thankful for editor’s comments and are pleased to submit the revised research manuscript titled “Exploring acculturative stress and coping mechanisms among pregnant South Asian immigrants in Ontario, Canada: A qualitative study protocol” (PONE-D-26-01077).

A point-by-point response is provided in a separate document and attached as ‘response to reviewers’.

The current word count of the manuscript is 3,304 (excluding title page, abstract, list of abbreviations, competing interests, authors’ contributions, acknowledgements, references).

Kindly note that we do not have any provision for further English language editing of this paper and request the Journal for the same.

This research is funded by the Social Sciences and Humanities Research Council Partnership Engage Grant (Application No. 892-2024-3059). The funders had no role in study design, data collection and analysis, decision to publish, or preparation of the manuscript.

We look forward to a publication of our manuscript in PLOS One

Sincerely,

Anam Shahil-Feroz (she/her)

Assistant Professor

Arthur Labatt Family School of Nursing, Faculty of Health Sciences

FIMS & Nursing Building Room 3338

Western University, London ON Canada N6A 5B9

https://www.uwo.ca/fhs/nursing/about/faculty/research_supervisors/anam_s.html

---

## [Editor Report · Decision Letter 1]

13 Mar 2026

Exploring acculturative stress and coping mechanisms among pregnant South Asian immigrants in Ontario, Canada: A qualitative study protocol

PONE-D-26-01077R1

Dear Dr. Shahil-Feroz,

We’re pleased to inform you that your manuscript has been judged scientifically suitable for publication and will be formally accepted for publication once it meets all outstanding technical requirements.

Kind regards,

Ietza Bojorquez, Ph.D.

Academic Editor

PLOS One
---

## [Editor Report · Acceptance letter]

PONE-D-26-01077R1

PLOS One

Dear Dr. Shahil-Feroz,

I'm pleased to inform you that your manuscript has been deemed suitable for publication in PLOS One. Congratulations! Your manuscript is now being handed over to our production team.

Kind regards,

on behalf of

Dr Ietza Bojorquez

Academic Editor

PLOS One